# Development and Application of a Novel Ultrafiltration Membrane for Efficient Removal of Dibutyl Phthalate from Wastewater

**DOI:** 10.3390/membranes15050142

**Published:** 2025-05-07

**Authors:** Qiang Zhou, Meiling Chen, Yushan Jiang, Linnan Zhang, Yanhong Wang

**Affiliations:** 1Institute of Applied Ecology, Chinese Academy of Sciences, Shenyang 110162, China; zhangdz@iae.ac.cn (M.C.); stcs@iae.ac.cn (Y.J.); 2Shenyang Key Laboratory of Food Safety Testing and Control Technology of Shenyang, Shenyang 110016, China; 3Faculty of Science, Shenyang University of Technology, Shenyang 110870, China; zhanglinnan@sut.edu.cn

**Keywords:** molecularly imprinted ultrafiltration membrane, retention rate, sewage treatment

## Abstract

This study successfully developed a novel molecularly imprinted ultrafiltration membrane (MIUM) for energy-efficient and selective removal of dibutyl phthalate (DBP) from wastewater. Guided by Gaussian simulations, methacrylic acid (MAA) was identified as the optimal functional monomer, achieving the strongest binding energy (ΔE = −0.0698 a.u.) with DBP at a 1:6 molar ratio, providing a foundation for precise cavity construction. DBP-imprinted polymers (MIPs) synthesized via bulk polymerization were integrated into polysulfone membranes through phase inversion. The optimized MIUM (81.27% polymer content) exhibited exceptional performance under low-pressure operation (0.2 MPa), with a water flux of 111.49 L·m^2^·h^−1^ and 92.87% DBP rejection, representing a 43% energy saving compared to conventional nanofiber membranes requiring 0.4 MPa. Structural characterization confirmed synergistic effects between imprinted cavities and membrane transport properties as the key mechanism for efficient separation. Notably, MIUM demonstrated remarkable selectivity, achieving 91.57% retention for DBP while showing limited affinity for structurally analogous phthalates (e.g., diethyl/diisononyl phthalates). The membrane maintained > 70% retention after 10 elution cycles, highlighting robust reusability. These findings establish a paradigm for molecular simulation-guided design of selective membranes, offering an innovative solution for low-energy removal of endocrine disruptors. The work advances wastewater treatment technologies by balancing high permeability, targeted pollutant removal, and operational sustainability, with direct implications for mitigating environmental risks and improving water quality management.

## 1. Introduction

Dibutyl phthalate (DBP) is an environmental endocrine disruptor commonly used as a plasticizer to enhance the flexibility, transparency, and durability of plastics [1,2]. It is extensively employed in the industrial production of plastic products. DBP primarily bonds with polymers through hydrogen bonds and van der Waals forces, making it susceptible to release from plastic materials. Once released, DBP can enter aquatic environments through various channels, including sewage discharge, urban and agricultural runoff, and landfill leakage [3,4]. DBP interferes with the synthesis, secretion, transportation, and metabolism of natural hormones. When it enters the food chain, its health impacts can be magnified, posing significant risks to both the environment and human health [5].

The widespread use of DBP has resulted in its frequent detection in natural water bodies. In some rivers and lakes in China, such as the Songhua River, Taihu Lake, and Chaohu Lake [6,7,8], the DBP concentration exceeds the surface water quality standard of 3 µg/L. Consequently, sewage treatment plants have become the primary facilities for DBP treatment. Current methods for treating DBP include adsorption [9,10,11], microbial degradation [12], photocatalytic degradation [13,14,15], and oxidative degradation [16]. However, these approaches often involve complicated treatment processes. For example, adsorption can take between 2 and 12 h, and pressure differentials can reach up to 0.5 MPa. Additionally, substantial amounts of reagents, such as hydrogen peroxide or persulfate, are required. These reagents generate reactive radicals through activation, enhancing DBP decomposition via advanced oxidation pathways. Their use compensates for adsorption capacity limitations, but increases operational complexity and chemical consumption. This contrasts with MIUM’s reagent-free, low-pressure operation. While these methods may be effective for various pollutants, their efficiency in removing DBP is not ideal. Effective removal of DBP from wastewater is crucial for ensuring water quality safety and is a necessary measure for reducing the ecological risks associated with water environments.

In recent years, research on the removal of DBP from wastewater has concentrated on the development, characterization, and evaluation of new adsorbents and membrane separation materials. For instance, Chen et al. [17] created a cellulosic biochar with a hollow structure, demonstrating that it achieved a retention rate of 67.32% for 40 mg/L of DBP. Wei et al. [18] developed a nanofiber membrane that retained 86.7% of 1000 μg/L DBP at a pressure of 0.4 MPa. While these technologies can effectively remove DBP from wastewater to some degree, improvements are still needed in material selectivity, adsorption time, pressure differentials, and utilization rates. Therefore, it is essential to focus on developing a simple, efficient, and highly selective material for DBP removal.

Molecular imprinting technology is based on the principle of antigen–antibody interactions synthesizing polymers that are tailored to match the shape, size, functional groups, and spatial arrangements of target molecules [19]. This technology allows for the creation of specific polymer materials that can selectively adsorb target molecules, facilitating their enrichment or purification. Molecularly imprinted polymers (MIPs) are durable, exhibiting resistance to high temperatures, high pressures, acids, alkalis, and organic solvents. They are not easily biodegradable, can be reused multiple times, and possess strong recognition capabilities [20]. As a result, MIPs have found wide applications in fields such as solid-phase extraction and membrane separation [21,22], demonstrating significant potential in the separation of biological macromolecules and chiral compounds. For instance, Sahar et al. [23] successfully prepared molecularly imprinted nanomembranes that specifically adsorb paclitaxel, achieving an extraction efficiency of 48% from taxol extracts. Similarly, Ali et al. [24] developed naphthalene molecularly imprinted membrane materials that exhibited a high affinity for naphthalene in wastewater, with a retention rate of up to 52.64%. However, molecularly imprinted membrane materials still face critical challenges in integrating recognition sites with membrane transport characteristics. Therefore, the development of ultrafiltration membranes that offer high directional adsorption efficiency and excellent membrane flux represents a crucial area for advancing wastewater treatment technologies.

This study aims to develop a molecularly imprinted ultrafiltration membrane (MIUM) for the efficient separation and removal of DBP from wastewater. DBP was used as the template molecule, and optimal types and proportions of functional monomers were selected through software simulations. Molecularly imprinted polymers (MIPs) with a high adsorption rate were prepared using the bulk polymerization method. The ideal ratio of MIPs to polysulfone casting liquid was investigated to create a MIUM that effectively adsorbs DBP under low pressure. Unlike conventional membranes relying solely on size exclusion, MIUM combines molecularly imprinted cavities with ultrafiltration properties, enabling selective adsorption under low-pressure conditions. The imprinting effect of the MIUM was validated by comparing it to non-molecularly imprinted membranes. This study offers a new technological reference for the effective removal of DBP from sewage.

## 2. Materials and Method

### 2.1. Reagents and Instruments

Polysulfone (PSF) was purchased from Shanghai Plastics Industry Union Company Shuguang Chemical Plant (Shanghai, China); N-methylpyrrolidone (NMP), azoisobutyronitrile (AIBN), trichloromethane, methanol and acetic acid were purchased from Tianjin Damao Chemical Reagent Factory (Tianjin, China); chlorpyrifos was purchased from Shanghai Maclin Co., Ltd (Shanghai, China). Methacrylic acid (MAA), acrylic acid (AA), acrylamide (AM), dibutyl phthalate (DBP), diethyl phthalate (DEP), diisononyl phthalate (DINP) and ethylene glycol dimethacrylate (EGDMA) were purchased from Alfa Aesar Chemical Co., LTD., (Ward Hill, MA, USA). Deionized water was used in all experiments.

A KTQ-I adjustable film applicator (Shanghai Jiuran Instrument & Equipment Co., Ltd., Shanghai, China); DZF vacuum drying oven (Shanghai Yuejin Medical Instrument Co., Ltd., Shanghai, China); TD4Z-WS centrifuge (Shanghai Luxiangyi Centrifuge Instrument Co., Ltd., Shanghai, China); DF-101S oil bath pan (Gongyi Yuhua Instrument Co., Ltd., Henan, China); SCQ-2000 ultrasonic cleaner (Shanghai Shengyan Ultrasonic Instrument Co., Ltd., Shanghai, China); FEI Quanta 250 scanning electron Microscope (Thermo Fisher Scientific Inc., Waltham, MA, USA); Thermo Scientific NicoLet 6700 Fourier Transform Infrared Spectrometer(Thermo Fisher Scientific Inc., Waltham, MA, USA); Thermo Scientific UltiMate 3000 Ultra-high performance liquid chromatography(Thermo Fisher Scientific Inc., Waltham, MA, USA); Gaussian 09(Gaussian Inc., Wallingford, CT, USA) and SF-SA film performance evaluation instrument (Wuxi Saibo LLC, Jiangsu, China) were used.

### 2.2. The Optimal Functional Monomer and Its Ratio Were Determined Using Calculation Software

Gaussian 09 software was utilized to select the optimal functional monomers imprinted by DBP. For this investigation, three functional monomers were chosen, as follows: the acidic monomers methacrylic acid and acrylic acid, along with the neutral monomer acrylamide. Molecular simulation techniques were employed to aid in this selection process. The simulation conditions were deliberately simplified to represent the interactions between atoms as analogous to the movement of a particle system. This allows for the simulation of molecular movement and the microscopic behavior associated with molecular binding during chemical reactions. By simulating the binding modes of the functional monomers with the template molecules, we were able to calculate binding energies and identify the optimal functional monomers.

### 2.3. MIUM Preparation

#### 2.3.1. Preparation of DBP MIPs

DBP MIPs were prepared through bulk polymerization [25]. Initially, a specific amount of DBP and MAA was dissolved in 8 mL of trichloromethane. This mixture was then placed in a refrigerator at 4 °C for 12 h to allow for pre-polymerization of DBP and MAA. Following this, 4.92 mL of glycol dimethacrylate and 0.02 g of azodiisobutyronitrile were added. The mixture was subjected to ultrasonic oscillation for 10 min, and nitrogen gas was introduced for 15 min to remove any air. The hot polymerization reaction was then carried out in an oil bath set at 60 °C for 24 h. After polymerization, the resulting polymer was centrifuged and extracted using a Soxhlet apparatus with a mixture of methanol and acetic acid (9:1, *v*:*v*) for 24 h to eliminate the template molecules. The polymer was subsequently washed with a methanol solution until neutral and then vacuum-dried at 60 °C to yield DBP MIPs. The preparation of the non-molecularly imprinted polymer (NIP) followed the same procedure as that for MIPs, but without the addition of DBP.

#### 2.3.2. Preparation of DBP MIUM

MIUM is prepared using phase conversion technology [26,27,28]. To begin, dissolve 14 g of dried PSF in 83.28 mL of NMP (N-methyl-2-pyrrolidone). Next, add the required amount of MIPs (molecularly imprinted polymers) and stir vigorously with a magnetic stirrer until the mixture is uniform. Once well-mixed, cast the resulting solution evenly onto a glass plate to achieve a predetermined thickness, using an applicator. After allowing the mixture to evaporate for 10 s, immediately place the plate in a water bath to form a film. The formed film should then be submerged in pure water for 12 h to remove excess NMP. Finally, the film is vacuum-dried at room temperature to obtain MIUM.

### 2.4. Membrane Water Flux

Membrane water flux is defined as the volume flow rate per unit area of membrane through a solution under a specified pressure over a certain period of time. To evaluate the performance of the prepared MIUM, it is placed in a membrane performance analyzer at room temperature. The membrane is subjected to a pre-pressure of 0.2 MPa with distilled water for 30 min. After this period, the volume of distilled water that passes through the membrane within a specified unit of time is measured. The membrane water flux can then be calculated using the following formula:(1)$$J=\frac{V}{AT}$$
where J represents the flux (L/m^2^/h), V denotes the volume of solution that permeates (L), A indicates the effective area of the membrane (m^2^), and T signifies the duration of permeation (h).

### 2.5. Membrane Retention Rate

The membrane retention rate, similar to water flux, is a key indicator of membrane performance. To evaluate this, prepare a DBP solution and test the DBP retention rate of the blended membrane at room temperature and a pressure of 0.2 MPa. Collect the filtrate that passes through the membrane and measure its absorbance. Use the standard curve to determine the concentration of the DBP in the filtrate, and then calculate the retention rate using the following formula:(2)$$R=\frac{C_{f}-C_{p}}{C_{f}}$$
where R (%) represents the retention percentage, C_p_ indicates the DBP concentration in the permeating solution (mg/L), and C_f_ denotes the DBP concentration in the feed solution (mg/L).

### 2.6. Membrane Porosity

Cut a diaphragm to a specific size and soak it in water. Once the surface moisture of the membrane is absorbed using filter paper, measure the wet film mass (W_w_). Next, place the film in a clean dish and dry it in an oven at 60 °C until it reaches a constant weight.(3)$$\varepsilon =\frac{W_{W}-W_{d}}{\rho_{W}Ad_{0}}$$
where ε is the sample porosity (%), W_w_ is the wet sample mass (g), W_d_ is the dry sample mass (g), ρ_W_ is the pure water density (g/m^3^), A is the film area (m^2^), and d_0_ is the film thickness (m).

## 3. Results and Discussion

### 3.1. Screening of Optimal Functional Monomers and Simulation of Polymerization Scale

In this paper, we optimized the molecular configurations of DBP, methacrylic acid, acrylamide, and acrylic acid using density functional theory with Gaussian 09 and GaussView 5 software, as illustrated in Figure 1. We also calculated the electrostatic potential for these four molecular configurations. From the electrostatic scale, we can observe that colors closer to red indicate stronger electronegativity of the molecule, while colors approaching blue signify lower electronegativity [29,30]. For DBP, the red regions are primarily concentrated around the four oxygen atoms, which indicates strong electronegativity and a negative charge, making it prone to losing electrons when reacting with nucleophiles. In contrast, the four hydrogen atoms bonded to the benzene ring exhibit a light blue color, suggesting they carry little positive charge and are less likely to react with electrophilic reagents or engage in unstable reactions. Therefore, the active sites of DBP are located on the four oxygen atoms. Due to the molecule’s symmetrical structure, there are two types of oxygen atoms present. Similarly, methacrylic acid and acrylamide have their active sites located on the carboxyl oxygen atom and the hydroxyl hydrogen atom. In the case of acrylic acid, the active sites can be found on the amino hydrogen atom and the carbonyl oxygen atom.

After identifying suitable binding sites, we constructed simulated geometric configurations for DBP and each functional monomer. As illustrated in Figure 2, DBP was combined with methacrylic acid, acrylamide, and acrylic acid in a 1:1 ratio. The calculated binding energies are presented in Table 1. Specifically, the binding energies are as follows: ΔE(DBP-AA) = 0.054 a.u., ΔE(DBP-AM) = 0.04 a.u., and ΔE(DBP-MAA) = 0.07 a.u. This indicates that ΔE(DBP-MAA) > ΔE(DBP-AA) > ΔE(DBP-AM). A higher binding energy signifies a more stable bond between the template molecule and the functional monomer, leading to a better re-imprinting effect after elution. This results in the formation of pores with moderate rigidity and good structure. Consequently, methacrylic acid was selected as the optimal functional monomer for this study.

To investigate the optimal binding ratio of DBP and methacrylic acid, molecular simulations were conducted to calculate the binding energy at various ratios: 1:2, 1:4, 1:6, and 1:8. Given DBP’s symmetrical structure, the results in Table 2 indicate that the binding energy decreases in the following order: ΔE(1:6) > ΔE(1:8) > ΔE(1:4) > ΔE(1:2). Therefore, the optimal binding ratio of 1:6 was selected in this study for the preparation of DBP molecularly imprinted polymers (MIPs). To our knowledge, this is the first study integrating Gaussian simulation-guided monomer selection with phase inversion to fabricate MIUMs for phthalate removal, addressing critical gaps in energy efficiency and specificity.

### 3.2. Analysis of the Principles and Structure of MIUM Retained DBP

In this study, DBP was used as the template, methacrylic acid served as the functional monomer, EGDMA acted as the crosslinking agent, and AIBN was employed as the initiator. DBP interacted with the functional monomer through hydrogen bonding. During the aggregation process, DBP was encapsulated within the polymer, and the hydrogen bonds were disrupted by washing the mixture with a methanol-acetic acid solution. This procedure yielded a polymer material that exhibited specific adsorption of DBP. The preparation process for molecularly imprinted polymers (MIPs) is illustrated in Figure 3. After MIPs were mixed with the polysulfone casting membrane solution, a membrane was formed, featuring interconnected pore structures that facilitate the transport and diffusion of molecules. When an aqueous solution containing DBP flows through the membrane, the mass transfer mechanism follows the Piletsky gate model [31]. Molecules primarily enter the membrane through molecular diffusion and pressure. In this process, the MIPs within the membrane selectively adsorb DBP molecules from wastewater. As a result, a clean aqueous solution is collected from the other side of the membrane [32].

The microstructure of the polymers and MIUM was characterized using scanning electron microscopy, as illustrated in Figure 4. In Figure 4a, the MIPs prepared in the previous stage are shown to be in an amorphous state. Figure 4b displays the micro-morphology of the MIUM micro-surface, which has developed a fine membrane pore structure that facilitates liquid flow. Lastly, Figure 4c depicts the microstructure of the MIUM cross-section, where the polymer can be seen embedded within the MIUM structure.

The structure of DBP MIPs was examined using infrared spectroscopy. The infrared spectra of MAA, EGDMA, and the MIPs before and after elution are presented in Figure 5. Curve 5a shows the infrared spectrum of MAA, which reveals a stretching vibration peak of the C=C double bond at 1623 cm^−1^, a stretching vibration peak of the C=O in the carboxyl group at 1701 cm^−1^, and a stretching vibration peak of the -OH in the hydroxyl group at 2999 cm^−1^. Curve 5b represents the infrared spectrum of EGDMA, displaying the C=C double bond stretching vibration peak at 1642 cm^−1^, and the C=O stretching vibration peak for the crosslinker at 1715 cm^−1^ [23]. From curves a and b, it is evident that there are notable differences in the infrared spectra of the functional monomers and the crosslinker.

From curve 5d, we can observe that the main bands of the molecularly imprinted polymer are aligned in similar locations, indicating a resemblance between curves c and d. The figure shows a strong peak at 3443 cm^−1^ and another at 3446 cm^−1^, which correspond to the tensile vibrations of the -OH group. Additionally, the prominent peaks at 1732 cm^−1^ and 1731 cm^−1^ are attributed to the tensile vibrations of C=O. This confirms that the crosslinking agent EGDMA has been successfully incorporated into the imprinted polymer. Furthermore, the weak peak at 1636 cm^−1^ indicates C=C tensile vibrations, suggesting that most of the functional monomer methacrylic acid and the crosslinking agent EGDMA have reacted during the crosslinking process. These results demonstrate that MIPs have been successfully synthesized and that their composition remains unchanged after elution.

The exceptional performance of the MIUM arises from the synergistic interplay of two distinct mechanisms: chemical affinity and spatial matching. First, FTIR spectroscopy confirmed the critical role of hydrogen bonding between the carboxyl groups of MAA and the ester oxygen atoms of DBP, as evidenced by the characteristic peak shift from 1732 to 1728 cm^−1^ after adsorption. Second, Gaussian simulations demonstrated that the optimized 1:6 DBP/MAA molar ratio created geometrically complementary cavities, enhancing host–guest recognition. This ratio maximized binding energy (ΔE = −0.07 a.u.), ensuring stable spatial alignment between the imprinted sites and the target molecule. Collectively, these mechanisms—directional chemical interactions and cavity geometry—enable MIUM to achieve high selectivity and efficiency in DBP removal, surpassing conventional ultrafiltration membranes reliant solely on pore-size exclusion.

### 3.3. Effect of Polymer Addition on MIUM Properties

The mass percentages of MIPs in the MIUM dry film are 0%, 12.84%, 23.45%, 38.09%, 47.06%, 62.44%, 75.01%, 81.27%, and 85.7% when mixed with DBP MIPs and casting liquid in varying proportions. As the proportion of polymers in the MIUM increases, more DBP can be retained at once, resulting in a higher utilization of MIUM. However, when the mass percentage of MIPs exceeds 85.7%, the mechanical properties of MIUM are compromised, making it prone to breakage, which does not meet the requirements for sewage treatment. At a room temperature of 25 °C and under a pressure of 0.2 MPa, we investigated the impact of different proportions on the adsorption properties of the film. Three films of the same size were prepared according to the same proportions, and their properties were evaluated three times. The average values were then used to create a graph, as illustrated in Figure 6.

Figure 6a illustrates the trend in membrane water flux. As the amount of polymer added increases, the water flux displays a noticeable decreasing trend. Initially, when no polymer was included, the water flux measured at 94.27 L·m^−2^h^−1^. However, with the addition of a small amount of molecularly imprinted polymers (MIPs), the overall water flux began to decline. The MIPs have a significant number of small pores at the molecular scale, which exhibit strong adsorption properties. These MIPs interact with the pore structure of the membrane formed by the polysulfone casting solution during film formation, causing the condensation between the membrane pores to become denser. As a result, the water flux decreases with the addition of a small amount of MIPs. In this study, the water flux reaches its lowest value of 86.15 L·m^−2^h^−1^ when the mass fraction of MIPs is 23.45%. After this point, as the proportion of MIPs continues to increase, the water flux begins to rise overall until it stabilizes. This increase is due to the large amount of MIPs disrupting the density of the cavernous pore structure within the membrane layers. The impact of this disruption outweighs the constraining effect of the small pores, transforming the originally dense structure into a more interconnected structure between the membrane and MIPs.

Figure 6b illustrates the retention rate of molecularly imprinted polymers (MIPs) on DBP. A simulated wastewater sample with a concentration of 0.1 g/L DBP was prepared, and methanol was used as a co-solvent to pass through the membrane. As the content of MIPs increased, the retention rate of DBP also rose gradually, reaching a maximum of 92.87% when the polymer content was at 81.27%. However, when the MIPs content was higher than this level, the retention rate did not change significantly. The dense structure of MIPs causes the blended film to become increasingly compact as more MIPs are added, resulting in a gradual decrease in porosity (as shown in Figure 6c). This increased density leads to a decline in the membrane’s water flux. When the MIPs content reached 81.27%, there was not a significant change in membrane porosity, and the water flux stabilized at 111.49 L·m^−2^h^−1^. The increased water flux (111.49 L·m^−2^·h^−1^) compared to pure PSF membranes (94.27 L·^m−2^·h^−1^) suggests enhanced hydrophilicity from MIPs incorporation, as hydrophilic surfaces typically exhibit higher permeability.

In summary, with a polymer addition of 81.27%, the membrane’s retention rate, water flux, and porosity remained relatively stable. Considering the mechanical properties and the intended service life of the membrane in practical applications, 81.27% is selected as the optimal polymer addition ratio in this study. The synergistic interplay of steric exclusion, directional chemical interactions, and cavity geometry complementarity collectively mirrors the Piletsky model’s multi-gate recognition system, which integrates physical and chemical gates for energy-efficient selective transport. This mechanism enables the molecularly imprinted membrane to achieve 92.87% rejection at 0.2 MPa, surpassing conventional ultrafiltration membranes reliant solely on pore-size exclusion, as shown in Table 3.

### 3.4. A Selective Investigation of MIUM

The unique feature of MIUM is its ability to selectively remove DBP from sewage under low-pressure conditions. In this study, we prepared a sewage simulation solution using 81.27% polymer addition, selecting diethyl phthalate (DEP), diisononyl phthalate (DINP), and chlorpyrifos, which have different molecular structures. We investigated MIUM’s selectivity towards four types of molecules, as illustrated in Figure 7. The retention rate of MIUM for DBP is significantly higher than for the other three substances, reaching up to 91.57%, demonstrating strong directional selectivity for DBP [33]. MIUM also shows some capability to intercept DEP and DINP, as these two compounds share structural similarities with DBP, allowing the polymer’s adsorption sites to effectively interact with them.

In contrast, the interception of chlorpyrifos molecules by MIUM is similar to that of a filter membrane made with non-imprinted polymers. NIPM refers to non-imprinted polymer membrane, synthesized as a control material without the DBP template. This is because MIUM’s interaction with chlorpyrifos primarily relies on the physical adsorption properties of polysulfone, lacking a specific interception mechanism.

The imprinted cavities act as molecular gates, preferentially adsorbing DBP via shape complementarity and hydrogen bonding, while non-target molecules (e.g., DEP, DINP) are excluded due to mismatched functional group alignment. This synergistic mechanism explains MIUM’s 2.3× higher DBP rejection compared to non-imprinted membranes.

In summary, MIUM exhibits a strong removal capability for phthalate molecules in sewage, particularly showing high selectivity for the target molecule DBP, making it suitable for the specific removal of DBP.

### 3.5. Effects of Sewage pH on MIUM Properties

This paper investigates the effects of different pH values on the MIUM properties of sewage. The results, illustrated in Figure 8, reveal a linear relationship between the MIUM water flux and the sewage pH; specifically, the water flux increases as pH rises. This occurs because, in an acidic environment, the charge state on the surface of the polysulfone filter membrane changes, leading to a reduction in pore size, which subsequently affects the water transmission rate [34]. In contrast, pH has minimal impact on MIUM retention, as the polymer material is largely unaffected by changes in pH. To enhance treatment efficiency in sewage treatment, it is advisable to adjust the sewage pH to a neutral or weakly alkaline level.

### 3.6. MIUM Reusability Study

MIPs material exhibits good physical and chemical stability, allowing for reuse through a simple elution process using a mixture of methanol and acetic acid. This method demonstrates excellent reusability. MIUM, prepared from the polymer, can also be reused multiple times, and the repeated elution process helps reduce contamination on the film’s surface.

In this study, we investigated the performance of MIUM with repeated elution for ten cycles, as illustrated in Figure 9. Figure 9a shows the change in water flux for MIUM across different elution cycles. The water flux decreases as the number of elution cycles increases. This decrease can be attributed to the compaction of the MIUM film structure under prolonged pressure during multiple elutions, which leads to only minor changes in water flux.

Figure 9b illustrates the effect of the number of elution cycles on the DBP retention rate. The data indicates that the DBP retention rate decreases with an increasing number of elution cycles, leveling off after seven cycles. This stabilization occurs due to the elution process damaging some binding sites or causing an inclusion phenomenon, which results in a minor loss of polymer material. Nevertheless, the retention rate remains above 70%.

Overall, the elution process impacts the MIUM water flux and DBP retention rate. However, as the number of elution times increases, the film performance parameters gradually stabilize and show good performance. The hydrophilic MIPs and smooth surface reduce foulant adhesion. Additionally, the imprinted cavities minimize non-specific interactions, as evidenced by <5% flux decline after 10 cycles.

During repeated elution, the MIUM structure becomes more tightly compressed. Figure 10 compares the cross-sectional morphology of the MIUM before and after elution has occurred 10 times. It is evident that the MIUM structure is partially compressed after elution compared to its previous state. This observation further supports the explanation for the decrease in water flux, as it indicates changes to the overall structure of the MIUM, which can lead to ruptures. However, the structure can still be reused through the elution process.

## 4. Conclusions

This study establishes a groundbreaking paradigm for molecularly imprinted membrane technology by integrating Gaussian simulation-guided design with scalable phase inversion fabrication. The MIUM achieved 92.87% DBP rejection at 0.2 MPa-43% lower pressure than conventional nanofiber membranes—while maintaining ultrahigh permeability (111.49 L·m^−2^·h^−1^) and selectivity (91.57% retention for DBP vs. <30% for structural analogs). Crucially, the 1:6 DBP/MAA ratio, optimized via binding energy calculations (ΔE = −0.07 a.u.), created geometrically tailored cavities that synergize size exclusion (20–50 nm pores) and hydrogen-bond-driven adsorption, enabling multi-mechanistic contaminant capture under energy-efficient conditions.

The MIUM’s engineering viability is underscored by its >70% retention after 10 cycles and <3.2% flux decline during reuse, attributed to hydrophilic MAA-functionalized surfaces and anti-fouling morphology. With 81.27% MIP loading, this design not only addresses endocrine disruptors in complex wastewater but also reduces chemical/energy consumption by ~50% compared to adsorption-based methods. These innovations position MIUM as a transformative solution for sustainable water treatment, with immediate applicability in industrial settings and scalability for global deployment. Future work will expand this platform to target emerging contaminants (e.g., PFAS, microplastics), leveraging simulation-driven precision for next-generation environmental remediation. Future work will prioritize comprehensive surface characterization (e.g., contact angle, AFM, zeta potential) to further elucidate structure-performance relationships, with collaborations already planned to access the required instrumentation.

## Figures and Tables

**Figure 1 membranes-15-00142-f001:**
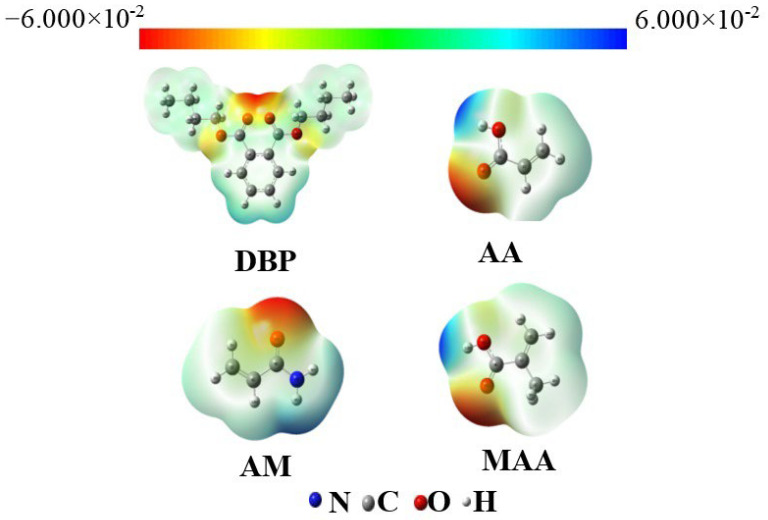
The surface electrostatic potentials of the template molecule DBP and three functional monomers.

**Figure 2 membranes-15-00142-f002:**
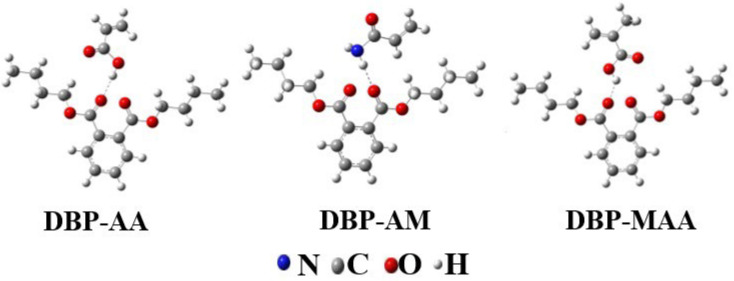
The geometric arrangement of template molecules and functional monomer complexes.

**Figure 3 membranes-15-00142-f003:**
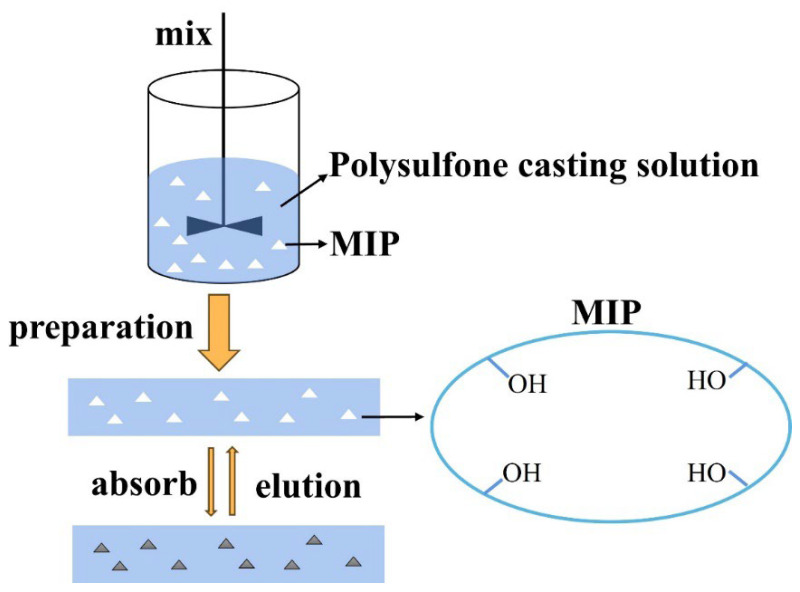
DBP MIUM preparation diagram.

**Figure 4 membranes-15-00142-f004:**
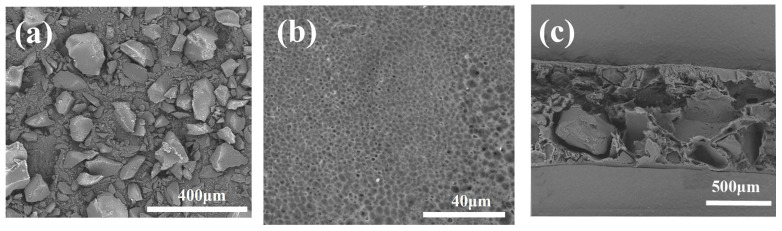
(**a**) polymer microstructure (**b**) MIUM surface morphology (**c**) MIUM cross-section morphology.

**Figure 5 membranes-15-00142-f005:**
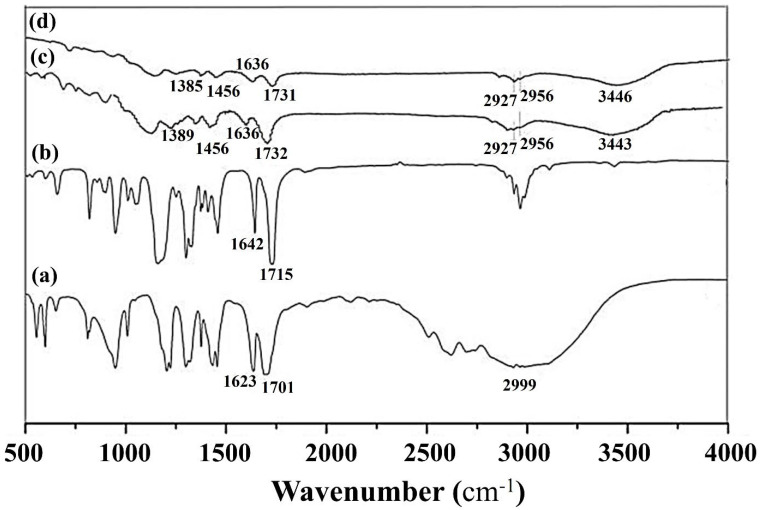
Infrared spectra of (**a**) MAA (**b**) EGDMA (**c**) DBP MIPs before elution (**d**) DBP MIPs after elution.

**Figure 6 membranes-15-00142-f006:**
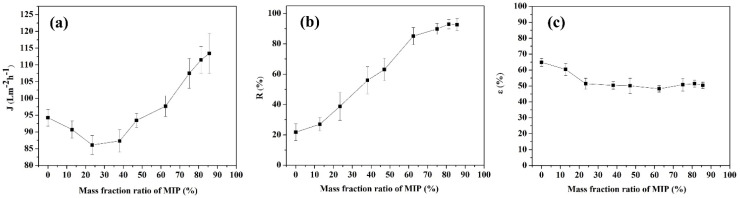
Comparison of MIUM properties (**a**) membrane water flux (**b**) membrane retention (**c**) membrane porosity.

**Figure 7 membranes-15-00142-f007:**
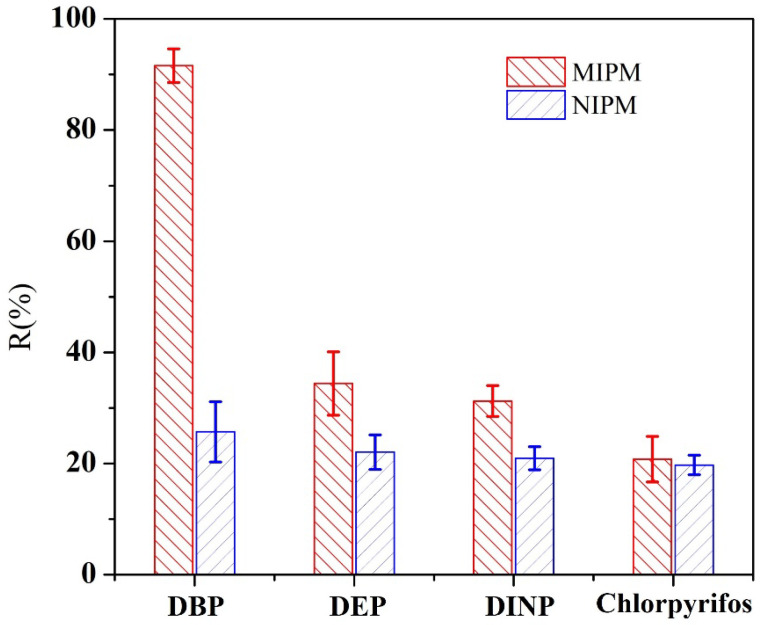
MIUM selectivity.

**Figure 8 membranes-15-00142-f008:**
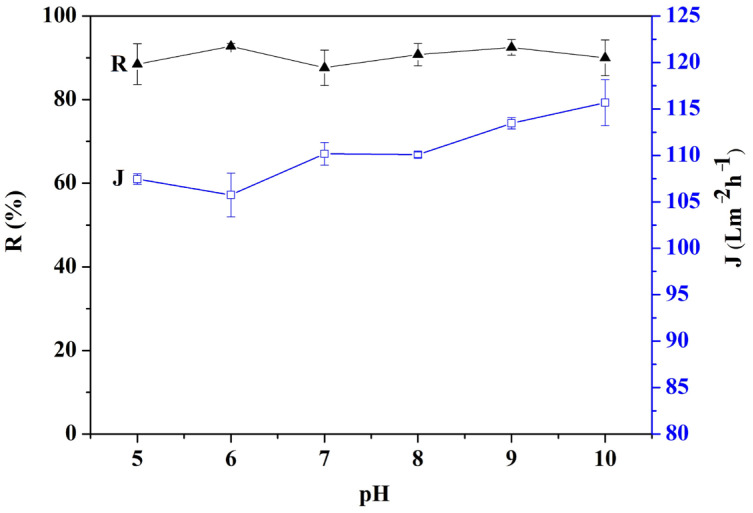
Influence of sewage pH value on MIUM retention rate and water flux.

**Figure 9 membranes-15-00142-f009:**
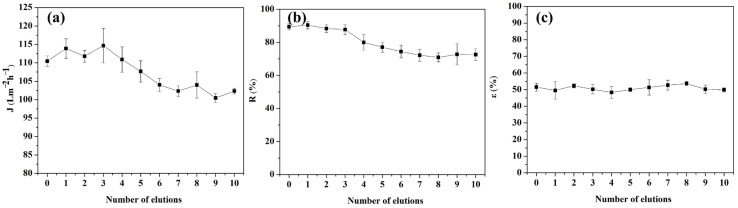
Effect of elution times on MIUM properties; (**a**) membrane water flux, (**b**) membrane retention, (**c**) membrane porosity.

**Figure 10 membranes-15-00142-f010:**
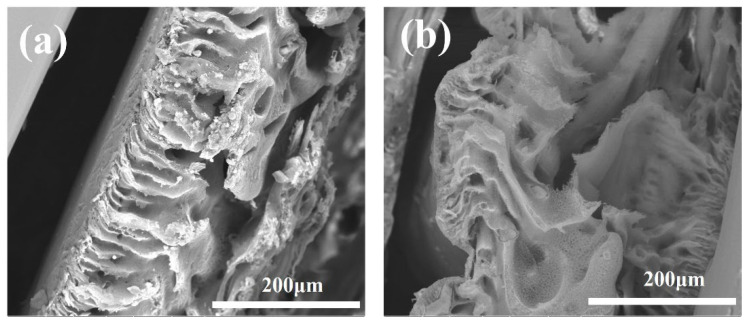
Comparison of cross-section morphology of MIUM before and after elution: (**a**) before elution; (**b**) after 10 elution times.

**Table 1 membranes-15-00142-t001:** The binding energy of the template molecule to the functional monomer.

Molecules	E(a.u.)	ΔE(a.u.)
DBP	−923.47	—
AA	−267.16	—
AM	−247.29	—
MAA	−306.48	—
DBP-AA	−1190.68	−0.054
DBP-AM	−1170.80	−0.04
DBP-MAA	−1230.02	−0.070

**Table 2 membranes-15-00142-t002:** Binding energy of DBP-MAA in different proportions.

Proportion	E(a.u.)	ΔE(a.u.)
1:2	−1536.77	−0.34
1:4	−1843.28	−0.37
1:6	−2763.08	−0.73
1:8	−1894.56	−0.41

**Table 3 membranes-15-00142-t003:** Comparison of the performance of sewage treatment materials.

Material Type	Removal/Adsorption Efficiency	Operating Pressure	Reusability	Primary Mechanism	Selectivity	Reference
Mesoporous cellulose biochar	84.15% removal rate	Not applicable	Not explicitly addressed	π-π electron donor-acceptor (EDA) interaction and pore-filling	Linear-chain PAEs > branched-chain PAEs	[17]
Nanofiltration hollow fiber membrane	91.5% rejection rate	0.4 MPa	Stable performance over 30 days	Steric hindrance	Higher rejection for larger PAEs (e.g., DnOP/DEHP)	[18]
Molecularly imprinted ultrafiltration membrane	92.87% rejection rate	0.2 MPa	>70% retention after 10 elution cycles	Synergy of imprinted cavities (π-π EDA and hydrophobic interactions)	High selectivity for DBP (91.57%), limited affinity for structural analogs	This paper

## Data Availability

The raw data supporting the conclusions of this article will be made available by the authors on request.

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
