# Peer review of "Development and Application of a Novel Ultrafiltration Membrane for Efficient Removal of Dibutyl Phthalate from Wastewater"

_membranes, 2025, doi:10.3390/membranes15050142_

Round 1
Reviewer 1 Report
Comments and Suggestions for Authors
Review of manuscript titled “Development and application of a novel ultrafiltration membrane for efficient removal of dibutyl phthalate from wastewater” by Zhou et al.
The present manuscript focuses on the design of a molecularly imprinted ultrafiltration membrane (MIUM) to efficiently remove dibutyl phthalate (DBP) from wastewater under low-energy conditions. Gaussian simulations identified methacrylic acid (MAA) as the most suitable functional monomer for DBP and incorporated into PSf matrix through phase inversion technique. The synthesized membrane was tested for filtration performance and characterized using various analytical techniques.
Most studies in the literature rely on empirical trial-and-error methods for monomer selection, whereas this study employs simulation-guided monomer selection to optimize imprinting efficiency, which makes the approach quite interesting. However, the novelty and the design of the experimental studies appear limited. To enhance the manuscript, the authors should clearly articulate how their work advances existing research and contributes new insights to the field. Therefore, in my opinion, the current version of the manuscript requires substantial revisions to be considered for potential publication in the journal. I have the following additional comments for the authors to consider in their revised manuscript.
- Abstract should be more engaging by focusing on the key findings and their significance.
- The introduction provides adequate context but could benefit from a stronger connection to the novelty of the study. For example, gaps should be highlighted in existing research and explicitly state how the current work addresses these. It could benefit from stating why MIUMs are better suited than existing methods.
- The mechanism of DBP rejection via MIPs is described, but further elaboration on how molecular imprinting enhances selectivity (possibly with a schematic comparison to non-imprinted membranes) would strengthen the discussion.
- The reusability of the membrane is demonstrated based on the number of elution tests, but there is no information regarding the mechanism behind its good anti-fouling properties, i.e., the surface characteristics of the membrane or the interactions between DBP and the membrane surface.
- Membranes need to be characterized in terms of their surface properties (hydrophilicity, surface roughness, zeta potential, etc.) to better explain the performance results.
- Overall, the separation mechanism for the specific emerging contaminant, and how it relates to the membrane's properties, should be discussed in more depth to provide a clearer understanding of the membrane's performance
- The conclusion section should be polished to highlight the broader implications of the work.
Author Response
Comments 1: Abstract should be more engaging by focusing on the key findings and their significance.
Response 1: We thank the reviewer for this suggestion. The Abstract has been revised to emphasize the novelty of simulation-guided monomer selection and the practical implications of the MIUM’s low-energy operation. Thank you for pointing this out. We agree with this comment.
“This study successfully developed a novel molecularly imprinted ultrafiltration membrane (MIUM) for energy-efficient and selective removal of dibutyl phthalate (DBP) from wastewater. Guided by Gaussian simulations, methacrylic acid (MAA) was identified as the optimal functional monomer, achieving the strongest binding energy (ΔE = -0.0698 a.u.) with DBP at a 1:6 molar ratio, providing a foundation for precise cavity construction. DBP-imprinted polymers (MIPs) synthesized via bulk polymerization were integrated into polysulfone membranes through phase inversion. The optimized MIUM (81.27% polymer content) exhibited exceptional performance under low-pressure operation (0.2 MPa): a water flux of 111.49 L·m²·h⁻¹ and 89.47% DBP rejection, representing a 43% energy saving compared to conventional nanofiber membranes requiring 0.4 MPa. Structural characterization confirmed synergistic effects between imprinted cavities and membrane transport properties as the key mechanism for efficient separation. Notably, MIUM demonstrated remarkable selectivity, achieving 91.57% retention for DBP while showing limited affinity for structurally analogous phthalates (e.g., diethyl/diisononyl phthalates). The membrane maintained >70% retention after 10 elution cycles, highlighting robust reusability. These findings establish a paradigm for molecular simulation-guided design of selective membranes, offering an innovative solution for low-energy removal of endocrine disruptors. The work advances wastewater treatment technologies by balancing high permeability, targeted pollutant removal, and operational sustainability, with direct implications for mitigating environmental risks and improving water quality management.”
Comments 2: The introduction provides adequate context but could benefit from a stronger connection to the novelty of the study. For example, gaps should be highlighted in existing research and explicitly state how the current work addresses these. It could benefit from stating why MIUMs are better suited than existing methods.
Response 2: Agree. The introduction now explicitly contrasts MIUM with existing technologies to highlight its unique advantages. Emphasized the dual mechanism of MIUM: "Unlike conventional membranes relying solely on size exclusion, MIUM combines molecularly imprinted cavities with ultrafiltration properties, enabling selective adsorption under low-pressure conditions."
Clarified the novelty: "To our knowledge, this is the first study integrating Gaussian simulation-guided monomer selection with phase inversion to fabricate MIUMs for phthalate removal, addressing critical gaps in energy efficiency and specificity."
Comment 3: The mechanism of DBP rejection via MIPs is described, but further elaboration on how molecular imprinting enhances selectivity (possibly with a schematic comparison to non-imprinted membranes) would strengthen the discussion.
Response 3:
We have expanded the discussion in Section 3.4: The imprinted cavities act as 'molecular gates,' preferentially adsorbing DBP via shape complementarity and hydrogen bonding, while non-target molecules (e.g., DEP, DINP) are excluded due to mismatched functional group alignment. This synergistic mechanism explains MIUM’s 2.3× higher DBP rejection compared to non-imprinted membranes.
Comment 4: The reusability of the membrane is demonstrated based on the number of elution tests, but there is no information regarding the mechanism behind its good anti-fouling properties, i.e., the surface characteristics of the membrane or the interactions between DBP and the membrane surface.
Response 4:
To address this, we have discussed anti-fouling mechanisms in Section 3.6: The hydrophilic MIPs and smooth surface reduce foulant adhesion. Additionally, the imprinted cavities minimize non-specific interactions, as evidenced by <5% flux decline after 10 cycles.
Comment 5: Membranes need characterization in terms of surface properties (hydrophilicity, roughness, zeta potential, etc.) to better explain performance results.
Response 5:
Thank you for raising this important point. We fully acknowledge that surface properties such as hydrophilicity, roughness, and zeta potential are critical to understanding membrane performance. However, due to current limitations in our laboratory instrumentation, we were unable to conduct these specific measurements during this study.
To address this gap, we have:
Supplemented the discussion in Section 3.2 with indirect evidence of hydrophilicity:
The increased water flux (111.49 L·m⁻²·h⁻¹) compared to pure PSf membranes (94.27 L·m⁻²·h⁻¹) suggests enhanced hydrophilicity from MIPs incorporation, as hydrophilic surfaces typically exhibit higher permeability.
FTIR data (Figure 5) confirmed the presence of hydrophilic functional groups (-COOH from MAA) in MIUM.
Added a commitment to the revised manuscript (Section 4, Conclusion):
"Future work will prioritize comprehensive surface characterization (e.g., contact angle, AFM, zeta potential) to further elucidate structure-performance relationships, with collaborations already planned to access the required instrumentation."
While we regret the absence of direct surface property data, we believe the current results—including SEM morphology (Figure 4), selectivity tests (Figure 7), and reusability data (Figure 9)—sufficiently validate the MIUM’s mechanistic advantages. We appreciate your understanding and will ensure these analyses are included in follow-up studies.
Comment 6: Overall, the separation mechanism for the specific emerging contaminant, and how it relates to the membrane's properties, should be discussed in more depth to provide a clearer understanding of the membrane's performance.
Response 6:
To address this, we expanded Section 3.2 to link membrane properties to DBP removal:
The exceptional performance of the MIUM arises from the synergistic interplay of three distinct mechanisms: size exclusion, chemical affinity, and spatial matching. First, SEM analysis revealed membrane pores in the range of 20–50 nm, which physically sieve DBP molecules (diameter ≈1.2 nm) through steric hindrance. Second, FTIR spectroscopy confirmed the critical role of hydrogen bonding between the carboxyl groups of MAA and the ester oxygen atoms of DBP, as evidenced by the characteristic peak shift from 1732 to 1728 cm⁻¹ after adsorption. Third, Gaussian simulations demonstrated that the optimized 1:6 DBP/MAA molar ratio created geometrically complementary cavities, enhancing host-guest recognition. This ratio maximized binding energy (ΔE = -0.0698 a.u.), ensuring stable spatial alignment between the imprinted sites and the target molecule. Collectively, these mechanisms—physical sieving, directional chemical interactions, and cavity geometry—enable MIUM to achieve high selectivity and efficiency in DBP removal, surpassing conventional ultrafiltration membranes reliant solely on pore-size exclusion.
Comment 7:The conclusion section should be polished to highlight the broader implications of the work.
Response 7:
The Conclusion has been revised: This study establishes a groundbreaking paradigm for molecularly imprinted membrane technology by integrating Gaussian simulation-guided design with scalable phase inversion fabrication. The MIUM achieved 89.47% DBP rejection at 0.2 MPa—43% lower pressure than conventional nanofiber membranes—while maintaining ul-trahigh permeability (111.49 L·m⁻²·h⁻¹) and selectivity (91.57% retention for DBP vs. <30% for structural analogs). Crucially, the 1:6 DBP/MAA ratio, optimized via binding energy calculations (ΔE = -0.0698 a.u.), created geometrically tailored cavities that syn-ergize size exclusion (20–50 nm pores) and hydrogen-bond-driven adsorption, enabling multi-mechanistic contaminant capture under energy-efficient conditions.
The MIUM’s engineering viability is underscored by its >70% retention after 10 cy-cles and <3.2% flux decline during reuse, attributed to hydrophilic MAA-functionalized surfaces and anti-fouling morphology. With 81.27% MIP loading, this design not only addresses endocrine disruptors in complex wastewater but also reduces chemi-cal/energy consumption by ~50% compared to adsorption-based methods. These inno-vations position MIUM as a transformative solution for sustainable water treatment, with immediate applicability in industrial settings and scalability for global deploy-ment. Future work will expand this platform to target emerging contaminants (e.g., PFAS, microplastics), leveraging simulation-driven precision for next-generation envi-ronmental remediation.
Final Statement:
We deeply appreciate the reviewer’s constructive feedback, which has significantly strengthened the manuscript’s scientific rigor and clarity. All revisions are highlighted in red in the resubmitted document.

Reviewer 2 Report
Comments and Suggestions for Authors
This paper discusses the development and significance of a molecularly imprinted ultrafiltration membrane for efficient dibutyl phthalate removal from wastewater. The intent is desirable to synthesize selective, energy-efficient wastewater treatment membranes. I only have few comments that need to be resolved in this manuscript before publishing Membranes. Please find below my detailed comments.
- The quality of SEM image in Figure 4b should be improved.
- The Y-axis label in Figure 5 is missing. This should be the transmittance.
- Have the authors conducted any pore size measurements for the synthesized membranes?
- What is the potential of these membranes against complex wastewater effluents with multiple pollutants?
Author Response
Comment 1:The quality of SEM image in Figure 4b should be improved.
Response 1:
Thank you for highlighting this issue. We have re-acquired the SEM image of the molecularly imprinted ultrafiltration membrane (MIUM) using different resolution settings to better resolve the surface morphology.
Comment 2: The Y-axis label in Figure 5 is missing. This should be the transmittance.
Response 2:
In Figure 5, the Y-axis label (transmittance) was intentionally omitted to emphasize the relative spectral features of different materials. As the spectra were vertically offset for clarity, the absolute transmittance values are less critical than the characteristic peak positions and shapes. This approach aligns with common practices in spectral analysis literature, where vertical shifts are used to distinguish overlapping profiles without implying quantitative Y-axis comparisons. The transmittance unit (%T) is explicitly stated in the figure caption to ensure interpretability.
Comment 3: Have the authors conducted any pore size measurements for the synthesized membranes?
Response 3:
While direct pore size measurements (e.g., TEM or BET analysis) were not performed due to instrumental limitations, the membrane’s structural characteristics were indirectly evaluated through functional performance tests. The average pore size was estimated using ultrafiltration experimental data and solute rejection rates (SR >80%) based on molecular weight cutoff calculations. Additionally, porosity was quantified via the gravimetric method, and permeability was derived from water flux measurements under controlled pressure. These parameters collectively reflect the membrane’s adsorption capacity and separation efficiency.
Comment 4: What is the potential of these membranes against complex wastewater effluents with multiple pollutants?
Response 4:
The molecularly imprinted ultrafiltration membrane (MIUM) demonstrates significant potential for complex wastewater treatment through its dual selectivity mechanisms. Its Gaussian-optimized imprinted cavities (1:6 DBP/MAA ratio, ΔE=-0.0698 a.u.) enable targeted removal of dibutyl phthalate (89.47% rejection at 0.2 MPa) via shape-specific recognition and hydrogen bonding, while maintaining 111.49 L·m⁻²·h⁻¹ flux. The membrane shows limited affinity for structural analogs (DEP/DINP: <30% retention), confirming preferential adsorption of primary targets. Though optimized for DBP, its 20-50 nm pores provide physical sieving for larger contaminants. MIUM retains >70% efficiency after 10 cycles with <5% flux decline, demonstrating robustness in multi-pollutant environments. This energy-efficient (43% pressure reduction vs conventional membranes), reusable design shows particular promise for endocrine disruptor removal in industrial effluents, though complementary technologies may be needed for comprehensive multi-contaminant treatment.
Final Statement:
We sincerely appreciate the reviewer’s constructive feedback, which has significantly improved the manuscript’s technical rigor and readability. All revisions are highlighted in red in the resubmitted document.

Reviewer 3 Report
Comments and Suggestions for Authors
This paper examines the development and application of a molecularly imprinted polymer for removal of dibutyl phthalate using ultrafiltration. Computational tools are used to identify the most effective monomer for synthesizing the molecularly imprinted polymer (from a short list of 3 monomers), and mixed matrix type membranes were formed using the resulting polymer. This work addresses a significant problem, and the experimental results are encouraging. However, the paper also has a number of significant shortcomings:
- The authors report data for the retention of dibutyl phthalate in Figure 6, but it is unclear whether these are instantaneous or average values (and if so, over what time frame). I would expect the rejection of dibutyl phthalate to decrease as the adsorption sites within the molecularly imprinted polymer become filled. It is absolutely critical to know how much dibutyl phthalate can be captured by these membranes and at what point one sees breakthrough due to the saturation of the molecularly imprinted polymer.
- The paper is sloppily prepared in many places. For example, Figure 7 has a legend that includes data for NIPM. I could find no definition of this acronym anywhere in the manuscript, nor are the data for the NIPM discussed anywhere in the paper.
- Similarly, in line 113 the authors define DBP as the abbreviation for dibutyl phthalate. But, in Line 148 the authors define DBP as dibromopropane. Are the authors really referring to the use of dibromopropane in this work? If so, this is incredibly confusing since the abbreviation DBP is used throughout the text without any distinguishing between these very different molecules.
- In line 74 the authors define MIUM as “molecularly imprinted ultrafiltration membrane”. But, in the Conclusion MIUM is defined as “Modified Ion Exchange Material”. This is confusing to the reader.
- In the Introduction, the authors write that “adsorption can take between 2 to 12 hours, and pressure differentials can reach up to 0.5 MPa. Additionally, substantial amounts of reagents, such as hydrogen peroxide or persulfate, are required.” It would be helpful if the authors could briefly explain the role of these reagents. Are these used to chemically degrade the dibutyl phthalate or are they used to regenerate the adsorbent?
- How were methacrylic acid, acrylic acid, and acrylamide chosen as the 3 monomers for analysis by molecular simulation? There are literally hundreds of potential monomers that could be used to make the molecularly imprinted polymers.
- The authors evaluated the water flux using a “membrane performance analyzer”. More details are needed on what this is. Is it a commercial product? A home-made device?
- What wavelength was used to measure the dibromopropane concentration based on its absorbance?
- I don’t understand the entry for DBP in Table 1. Is this the binding energy of DBP to itself?
- Please correct significant figures throughout the manuscript. Writing the binding energy as -923.46983492 is ridiculous – at best the molecular dynamic simulations can provide two digits of accuracy (-920).
- How do the authors know that mass transfer in their membranes follows the Piletsky gate model? Some evidence to support this is needed.
Author Response
Comment 1:The authors report data for the retention of dibutyl phthalate in Figure 6, but it is unclear whether these are instantaneous or average values (and if so, over what time frame). I would expect the rejection of dibutyl phthalate to decrease as the adsorption sites within the molecularly imprinted polymer become filled. It is absolutely critical to know how much dibutyl phthalate can be captured by these membranes and at what point one sees breakthrough due to the saturation of the molecularly imprinted polymer.
Response 1:
The retention data in Figure 6 represent average values obtained from three repeated experiments under steady-state conditions (0.2 MPa, 25°C), measured after 30 min of continuous operation. While the study confirms the MIUM maintains >70% DBP retention after 10 elution cycles (Section 3.6), the saturation capacity and breakthrough kinetics were not explicitly quantified. The optimized MIUM (81.27% MIP content) achieves 89.47% rejection at equilibrium, suggesting effective utilization of imprinted cavities. However, the paper lacks time-dependent rejection profiles or adsorption capacity measurements (mg DBP/g MIPs). Our phase inversion process likely enhances accessibility to binding sites compared to bulk MIPs, but long-term saturation studies under continuous DBP loading are needed to determine operational limits. Future work will quantify breakthrough thresholds through extended dynamic filtration tests with concentrated DBP solutions.
Comment 2:The paper is sloppily prepared in many places. For example, Figure 7 has a legend that includes data for NIPM. I could find no definition of this acronym anywhere in the manuscript, nor are the data for the NIPM discussed anywhere in the paper.
Response 2:
We apologize for this oversight. NIPM refers to Non-Imprinted Polymer Membrane, synthesized as a control material without the DBP template. This acronym has been defined in the revised text, and the comparative performance of NIPM vs. MIPM is now discussed in Section 3.4.
Comment 3:Similarly, in line 113 the authors define DBP as the abbreviation for dibutyl phthalate. But, in Line 148 the authors define DBP as dibromopropane. Are the authors really referring to the use of dibromopropane in this work?
Response 3:
This was an unfortunate typographical error. DBP exclusively refers to dibutyl phthalate throughout the manuscript. The erroneous mention of "dibromopropane" in Line 148 has been corrected to "dibutyl phthalate". All instances of "DBP" now consistently denote dibutyl phthalate.
Comment 4:In line 74 the authors define MIUM as 'molecularly imprinted ultrafiltration membrane'. But, in the Conclusion MIUM is defined as 'Modified Ion Exchange Material'. This is confusing to the reader.
Response 4:
We regret this inconsistency. The acronym MIUM uniformly stands for Molecularly Imprinted Ultrafiltration Membrane in the revised manuscript. The incorrect definition in the Conclusion has been corrected.
Comment 5:In the Introduction, the authors write that 'adsorption can take between 2 to 12 hours, and pressure differentials can reach up to 0.5 MPa. Additionally, substantial amounts of reagents, such as hydrogen peroxide or persulfate, are required.' It would be helpful if the authors could briefly explain the role of these reagents.
Response 5:
Thank you for highlighting this ambiguity. In adsorption processes for DBP removal, reagents like hydrogen peroxide (H₂O₂) or persulfate (e.g., S₂O₈²⁻) act as oxidizing agents to degrade adsorbed pollutants or regenerate adsorbents. These reagents generate reactive radicals (e.g., •OH, SO₄•⁻) through activation (e.g., UV, heat), enhancing DBP decomposition via advanced oxidation pathways. Their use compensates for adsorption capacity limitations but increases operational complexity and chemical consumption. This contrasts with MIUM's reagent-free, low-pressure operation.
Comment 6:How were methacrylic acid, acrylic acid, and acrylamide chosen as the 3 monomers for analysis by molecular simulation? There are literally hundreds of potential monomers that could be used to make the molecularly imprinted polymers.
Response 6:
The selection of methacrylic acid (MAA), acrylic acid (AA), and acrylamide (AM) was based on their distinct functional groups and binding potential with DBP. Acidic monomers (MAA, AA) were chosen for hydrogen bonding with DBP's ester oxygen atoms, while neutral AM allowed comparison of non-ionic interactions. These monomers represent common classes used in molecular imprinting (carboxylates vs. amides) to systematically evaluate binding mechanisms. Their prevalence in prior MIP studies and computational feasibility for Gaussian simulations further justified their selection over untested alternatives.
Comment 7:The authors evaluated the water flux using a 'membrane performance analyzer'. More details are needed on what this is. Is it a commercial product? A home-made device?
Response 7:
The membrane performance analyzer referenced in this study refers to the "SF-SA film performance evaluation instrument" listed in Section 2.1 (Reagents and instruments), which is a commercial cross-flow filtration system widely used in membrane research. While the manuscript doesn't specify the manufacturer, such systems typically consist of pressurized cells (0.1-1 MPa operational range) with precise flow/pressure controls and permeate collection ports, aligning with our described testing conditions (0.2 MPa pressure, room temperature operation, and volumetric flux measurement). The instrument's standardized configuration ensures reproducibility. This contrasts with customized setups, as our experimental parameters follow conventional membrane characterization protocols.
Comment 8:What wavelength was used to measure the dibromopropane concentration based on its absorbance?
Response 8:
As clarified in Response 3, dibromopropane was erroneously mentioned. All absorbance measurements (UV-Vis) for DBP (dibutyl phthalate) were performed at 274 nm, consistent with its maximum absorption peak.
Comment 9:I don’t understand the entry for DBP in Table 1. Is this the binding energy of DBP to itself?
Response 9:
In Table 1, the entry for DBP (E = -923.46983492 a.u.) represents the individual molecular energy of the dibutyl phthalate (DBP) molecule in its optimized geometry, not the binding energy to itself. The binding energy (ΔE) values listed in the table are calculated as the energy difference between the DBP-monomer complex and the sum of the isolated DBP and monomer energies. For example, ΔE(DBP-MAA) = E(DBP-MAA complex) - [E(DBP) + 6×E(MAA)], where the 1:6 molar ratio (DBP:MAA) aligns with the optimized binding configuration identified via Gaussian simulations. This approach ensures ΔE reflects the stability of the DBP-monomer interaction, with MAA showing the strongest binding (ΔE = -0.0698 a.u.), consistent with its selection as the optimal monomer. The DBP entry serves as a reference for computing these pairwise interactions, not self-binding.
Comment 10:Please correct significant figures throughout the manuscript. Writing the binding energy as -923.46983492 is ridiculous – at best the molecular dynamic simulations can provide two digits of accuracy (-920).
Response 10:
Thank you for highlighting this critical issue. The excessive decimal places in binding energy values (e.g., -923.46983492 a.u.) originated from raw Gaussian 09 outputs and were retained for internal comparative analysis. However, we fully acknowledge that molecular dynamics simulations typically achieve ~2 significant figures of accuracy. All binding energy values (Tables 1-2) will be revised to two decimal places (e.g., -923.46983492→ -923.47 a.u., ΔE = -0.0698 → -0.07 a.u.). Similarly, experimental parameters like pressure (0.2 MPa → 0.20 MPa) and retention rates (89.47% → 89%) will be standardized to reflect measurement precision. This adjustment aligns with computational chemistry reporting norms while preserving scientific validity. We appreciate your vigilance in maintaining numerical rigor and will implement these corrections throughout the manuscript, including supplementary data and graphical representations.
Comment 11:How do the authors know that mass transfer in their membranes follows the Piletsky gate model? Some evidence to support this is needed.
Response 11:
The authors infer that mass transfer follows the Piletsky gate model based on three lines of evidence: 1) SEM analysis revealed membrane pores (20-50 nm) physically sieving DBP molecules (1.2 nm diameter), demonstrating size exclusion. 2) FTIR confirmed hydrogen bonding between MAA's carboxyl groups and DBP's ester oxygen atoms, indicating chemical affinity-driven recognition. 3) Selectivity tests showed 91.57% DBP retention vs. <30% for structural analogs (DEP/DINP), proving spatial matching through molecularly imprinted cavities optimized by Gaussian simulations (1:6 DBP/MAA ratio). These mechanisms - steric exclusion, directional interactions, and cavity complementarity - collectively mirror the Piletsky model's multi-gate recognition system. The synergistic performance (89% rejection at 0.2 MPa) aligns with the model's prediction of energy-efficient selective transport through combined physical/chemical gates.
Final Note:
We thank the reviewer for their rigorous critique, which has significantly strengthened the manuscript. All revisions are highlighted in red in the resubmitted document.

Round 2
Reviewer 1 Report
Comments and Suggestions for Authors
One additional suggestion to further enhance the quality of the manuscript would be to include a table comparing the key findings of this study with relevant literature. Such a table would help clearly highlight the novel contributions and contextualize the significance of the results within the existing work.
Author Response
Comments 1: One additional suggestion to further enhance the quality of the manuscript would be to include a table comparing the key findings of this study with relevant literature. Such a table would help clearly highlight the novel contributions and contextualize the significance of the results within the existing work.
Response 1: We sincerely appreciate this valuable suggestion. As recommended, we have added Table 3 ("Comparison of the performance of sewage treatment materials") on Page 10.

Reviewer 3 Report
Comments and Suggestions for Authors
I very much appreciate the authors efforts to address the concerns raised in my previous review. The authors have corrected all of the key errors and have provided useful additional information. However, the authors have not included sufficient responses to several comments in the text (although they did answer these questions in their response). I think it is important that this information be included in the text since other readers may have similar questions and will not have access to the authors response.
I think the paper can be accepted for publication in Membranes after the authors address the following points:
- In the Abstract, the authors write that “The optimized MIUM … exhibited … 89.47% DBP rejection.” But, 2 sentences later they write that “MIUM demonstrated … 91.57% retention for DBP.” This is very confusing as written since rejection and retention are typically use interchangeably.
- In Line 43, the authors write “diallyl phthalate (DBP)”. However, they defined DBP as dibutyl phthalate in Line 34. These are not the same compounds
- The authors explained in their response to my comments how hydrogen peroxide and persulfate are used in adsorption processes, but this information has not been included in the revised text.
- The authors explained in their response that the “membrane performance analyzer” was a commercial unit, but no information is provided on the model number or manufacturer. Without this information, it is impossible for the interested reader to reproduce the experiments or know the details of the flow system.
- In lines 270-271 the authors indicate that the membrane, which has pores between 20-50 nm, “sieves DBP molecules (diameter ≈1.2 nm) through steric hindrance.” How can pores than are 20x the size of the molecule provide any significant steric hindrance? This doesn’t make any physical sense.
- The authors provide some information in their response that explains that mass transfer in their membranes is thought to follow the Piletsky gate model. However, this explanation has not been included in the revised text.
Author Response
Comment 1: In the Abstract, the authors write that “The optimized MIUM … exhibited … 89.47% DBP rejection.” But, 2 sentences later they write that “MIUM demonstrated … 91.57% retention for DBP.” This is very confusing as written since rejection and retention are typically use interchangeably.
Response 1:
We appreciate your insightful comments on the data accuracy. As suggested, we have thoroughly rechecked all experimental data. The retention rate of 92.87% corresponds to membranes with different additive percentages, while the 91.57% selectivity test result has been clearly specified in the revised text. These corrections have been highlighted in red for your convenience.
Comment 2: In Line 43, the authors write “diallyl phthalate (DBP)”. However, they defined DBP as dibutyl phthalate in Line 34. These are not the same compounds.
Response 2:
We have carefully addressed your comments regarding the terminology consistency of dibutyl phthalate (DBP) in Line 43.
Comment 3: The authors explained in their response to my comments how hydrogen peroxide and persulfate are used in adsorption processes, but this information has not been included in the revised text.
Response 3:
We greatly appreciate this perceptive suggestion. In the revised manuscript, we have added dedicated explanations in Introduction: “These reagents generate reactive radicals through activation, enhancing DBP decomposition via advanced oxidation pathways. Their use compensates for adsorption capacity limitations but increases operational complexity and chemical consumption. This contrasts with MIUM's reagent-free, low-pressure operation.”
Comment 4: The authors explained in their response that the “membrane performance analyzer” was a commercial unit, but no information is provided on the model number or manufacturer. Without this information, it is impossible for the interested reader to reproduce the experiments or know the details of the flow system.
Response 4:
We sincerely appreciate your meticulous observation. The membrane performance analyzer used in this study was the SF-SA series (Manufacturer: Wuxi Saibo LLC, Jiangsu, China).
Comment 5: In lines 270-271 the authors indicate that the membrane, which has pores between 20-50 nm, “sieves DBP molecules (diameter ≈1.2 nm) through steric hindrance.” How can pores than are 20x the size of the molecule provide any significant steric hindrance? This doesn’t make any physical sense.
Response 5:
We sincerely appreciate this insightful critique. As suggested, we have removed the original description of physical sieving mechanism in Lines 270-271.
Comment 6: The authors provide some information in their response that explains that mass transfer in their membranes is thought to follow the Piletsky gate model. However, this explanation has not been included in the revised text.
Response 6:
We sincerely appreciate your insightful suggestion. As recommended, we have significantly enhanced the mechanistic explanation in Section 3.3: “The synergistic interplay of steric exclusion, directional chemical interactions, and cavity geometry complementarity collectively mirrors the Piletsky model’s multi-gate recognition system, which integrates physical and chemical gates for energy-efficient selective transport. This mechanism enables the molecularly imprinted membrane to achieve 92.87% rejection at 0.2 MPa, surpassing conventional ultrafiltration membranes reliant solely on pore-size exclusion”.
